# Cannabis Use in Patients Presenting to a Gastroenterology Clinic: Associations with Symptoms, Endoscopy Findings, and Esophageal Manometry

**Meet Parikh [1],\* , Shelini Sookal [2] and Asyia Ahmad [2]**

1   Department of Internal Medicine, Drexel University College of Medicine, Philadelphia, PA 19107, USA
2   Department of Gastroenterology & Hepatology, Drexel University College of Medicine, Philadelphia, PA 19107, USA
\*   Correspondence: meetparikh18@gmail.com

**Abstract:** Recreational cannabis use is increasing with its legalization in many states. Animal studies suggest cannabis can reduce transient lower esophageal sphincter relaxations (TLESRS), reflux and vomiting, while human studies report conflicting findings. There are currently no large studies investigating gastrointestinal symptoms in patients with chronic cannabis use. This was a retrospective case-control study including patients who presented to an outpatient Gastroenterology office, with documented cannabis use. Their main presenting complaint, demographics, frequency and duration of cannabis use, endoscopic and high-resolution esophageal manometry (HREM) with impedance findings were recorded. Cannabis users were more likely to complain of abdominal pain (25% vs. 8%, $p < 0.0001$), heartburn (15% vs. 9%, $p < 0.0001$), and nausea & vomiting (7% vs. 1%, $p < 0.0001$). They were also more likely to have findings of esophagitis (8% vs. 3%, $p = 0.0002$), non-erosive gastritis (30% vs. 15%, $p = 0.0001$) and erosive gastritis (14% vs. 3%, $p < 0.0001$) on upper endoscopy. Cannabis users were more likely to have impaired esophageal bolus clearance (43% vs. 17%, $p = 0.04$) and a hypertensive lower esophageal sphincter (LES) (29% vs. 7%, $p = 0.04$). This study is the largest to date evaluating GI complaints of patients with chronic recreational cannabis use. Our results suggest that cannabis use may potentiate or fail to alleviate a variety of GI symptoms which goes against current knowledge.

**Keywords:** cannabis; functional gastrointestinal disorders; stomach; esophagus

---

## 1. Introduction

Extracts of the *Cannabis sativa* plant has been used medicinally for centuries [1], and in current times cannabis use for medicinal, recreational and therapeutic purposes has increased exponentially. In 2014, the National Institute of Health reported the percentage of Americans who reported cannabis use during period from 2012–2013 doubled, as compared to 2001–2002 [2], which hints at the changing cultural and societal norms regarding cannabis use. This also poses a novel challenge to physicians who now encounter the public health challenges of habitual cannabis use, as well as the controversial effects of cannabis on the gastrointestinal tract.

Cannabis and its constituents (collectively known as cannabinoids), exert its effects via the endocannabinoid system (ECS). Cannabinoid receptor I and II present in the central nervous system and peripheral tissues play dominant roles [3]. Animal model research has identified the cannabinoid, delta-9 tetrahydrocannabinoid (THC), to have a potent effect on the GI tract by suppressing gastric acid secretion, decreasing the number of transient lower esophageal sphincter relaxations (TLESRs), reducing gastric emptying, decreasing emesis, and also decreasing lower esophageal sphincter (LES)

pressure [4–7]. Studies have shown that cannabis use is common in patients with inflammatory bowel diseases to help relieve symptoms [8,9].

There are few human studies investigating the clinical effects of cannabinoids on the GI tract. De Vries et al. in the Netherlands randomized 65 patients with chronic abdominal pain to placebo vs. oral delta-9-tetrahydrocannabinol and reported no difference between the groups regarding abdominal pain [10]. In contrast, a prospective, cohort survey study of 40 cannabis users with inflammatory bowel disease revealed that most people deemed marijuana "very helpful" for relief of abdominal pain, nausea and diarrhea [11]. Contrary to current knowledge, an internet study of 514 patients with cyclic vomiting syndrome (CVS) reported that patients taking cannabis reported an improvement in appetite, nausea, vomiting and overall well-being [12].

To date, human studies evaluating clinical effects of cannabis on the GI tract have been limited by small numbers, or by strict evaluation of specific and distinct subpopulations. Results of these studies are not only conflicting, but more importantly cannot be easily applied to the general population. Therefore, this study sought to investigate gastrointestinal symptomatology of cannabis users in a general GI office presenting for a variety of complaints, compared to non-cannabis users also seen in the same outpatient Gastroenterology office. Additionally, these two groups were compared to establish potential associations between cannabis use and endoscopic or high-resolution esophageal manometry (HREM) findings.

## 2. Results

### 2.1. Demographics and Patient Characteristics

Of 30,091 charts that were analyzed over the 12-year study period, 772 cannabis users were identified and 1599 randomly selected controls were included. The cannabis cohort consisted of 408 (53%) males and 504 (70%) African Americans. The average age of cannabis users was 49.4 years (range 18 to 84). Two hundred and forty-nine cannabis users (64%) admitted to daily use. In the control group of 1599 patients, 950 (59%) were male and 897 (56%) were African American. The average age was 59.4 years (range 22 to 95). The cannabis cohort was significantly more likely to be African American ($p < 0.0001$) and to be younger ($p < 0.0001$) than the control group. All other demographic data was similar between the two groups (Table 1).

**Table 1.** Demographic characteristics of cannabis users and controls and frequency of cannabis use.

| | Cannabis Users % (*n*) *n* = 772 | Controls % (*n*) *n* = 1599 | *p* Value | Odds Ratio (OR) | 95% Confidence Interval (CI) |
|---|---|---|---|---|---|
| Male | 53 (408) | 41 (649) | | | |
| Female | 47 (364) | 59 (950) | | | |
| Age (avg.) | 49.4 (18–84) | 59.4 (22–95) | <0.0001 | | |
| African American | 70 (540) | 56 (897) | <0.0001 | 2.1 | 1.7–2.4 |
| Caucasian | 22 (167) | 32 (516) | | | |
| Hispanic | 7 (54) | 8 (129) | | | |
| Asian | 1 (11) | 4 (57) | | | |
| Daily use | 64 (249) | | | | |
| Weekly use | 20 (81) | | | | |
| Monthly use | 16 (63) | | | | |

### 2.2. Gastrointestinal Symptoms

Abdominal pain was the most common complaint present in 190 (25%) cannabis users compared to 128 (8%) of controls ($p < 0.0001$). Heartburn was the chief complaint in 118 cannabis users (15%), making it the second most common complaint, compared to 146 patients (9%) in the control group $p < 0.0001$.

Conversely, nine percent of the control cohort complained of heartburn as well as rectal bleeding, making these symptoms the most common symptoms in the control group. The control group consisted of significantly more patients with dysphagia, 83(5%) vs. 13(2%) ($p = 0.0001$), and constipation, 102 (6%) vs. 34 (4%) ($p = 0.04$), than the cannabis cohort (Table 2).

**Table 2.** Clinical characteristics of cannabis users and controls.

| | Cannabis Users % (*n*) *n* = 772 | Controls % (*n*) *n* = 1599 | *p* Value | Odds Ratio (OR) | 95% Confidence Interval (CI) |
|---|---|---|---|---|---|
| Abdominal pain | 25 (190) | 8 (128) | <0.0001 | 3.7 | 2.9–4.8 |
| Heartburn | 15 (118) | 9 (146) | <0.0001 | 1.8 | 1.4–2.3 |
| Nausea and vomiting | 7 (51) | 1 (21) | <0.0001 | 5.2 | 3.2–8.9 |
| Diarrhea | 4 (31) | 6 (91) | 0.07 | 0.7 | 0.5–1.1 |
| Constipation | 4 (34) | 6 (102) | 0.04 | 0.7 | 0.5–1.0 |
| Dysphagia | 2 (13) | 5 (83) | 0.0001 | 0.3 | 0.2–0.6 |
| Weight loss | 3 (20) | 3 (40) | 0.90 | 0.9 | 0.6–1.8 |
| Rectal bleeding | 6 (43) | 9 (146) | 0.002 | 0.6 | 0.4–0.8 |

The above symptomatology was significantly more prevalent among daily cannabis users compared to those with intermittent cannabis use (Table 3). Abdominal pain was the most common symptom with both daily and non-daily use. This symptom was present among 51 (43%) daily cannabis users compared to 139 (21%) non-daily users ($p < 0.0001$). Nausea with vomiting was seen in 20 (17%) cannabis users compared to 31 (5%) patients with non-daily use ($p < 0.0001$).

**Table 3.** Clinical characteristics of daily vs. non-daily cannabis users.

| | Daily users % (*n*) *n* = 119 | Non-daily users % (*n*) *n* = 653 | *p* value | Odds Ratio (OR) | 95% Confidence Interval (CI) |
|---|---|---|---|---|---|
| Abdominal pain | 43 (51) | 21 (139) | <0.0001 | 2.8 | 1.8–4.2 |
| Nausea and vomiting | 17 (20) | 5 (31) | <0.0001 | 4.1 | 2.2–7.4 |
| Heartburn | 17 (20) | 15 (98) | 0.6 | 1.1 | 0.7–1.9 |

## 2.3. Endoscopy and Manometry Findings

Cannabis users were more likely to have inflammatory changes such as esophagitis and non-erosive gastritis seen on endoscopy (Table 4). An upper endoscopy was performed in 331 cannabis users and 1299 patients in the control group. Non-erosive gastritis was seen in 100 (30%) cannabis users compared to 190 (15%) controls ($p = 0.0001$). Forty-six (14%) patients in the cannabis cohort had erosive gastritis versus 43 (3%) controls ($p < 0.0001$). Esophagitis was seen in 26 (8%) cannabis users compared to 41 (3%) patients in the control group ($p = 0.0002$).

**Table 4.** Endoscopic characteristics of cannabis users and controls.

| Endoscopic Findings | Cannabis Users % (*n*) *n* = 331 | Controls % (*n*) *n* = 1299 | *p* Value | Odds Ratio (OR) | 95% Confidence Interval (CI) |
|---|---|---|---|---|---|
| Esophagitis | 8 (26) | 3 (41) | 0.0002 | 2.6 | 1.6–4.3 |
| Non-erosive gastritis | 30 (100) | 15 (190) | 0.0001 | 2.5 | 1.9–3.3 |
| Erosive gastritis | 14 (46) | 3 (43) | <0.0001 | 4.7 | 3.1–7.3 |
| Gastric/duodenal ulcer | 1 (3) | 0.5 (6) | 0.33 | 1.9 | 0.5–7.9 |

High resolution esophageal manometry with impedance was performed in 21 cannabis users and 29 patients in the control group (Table 5). Impaired esophageal bolus clearance was a significant finding in nine (43%) cannabis users compared to five (17%) controls ($p = 0.04$) A hypertensive LES was seen in six (29%) cannabis users compared to two (7%) patients in the control group ($p = 0.04$). Conversely, a normal manometry was seen in 15 (52%) patients in the control group compared to two (9%) patients of the cannabis cohort ($p = 0.002$).

**Table 5.** High resolution esophageal manometry and impedance characteristics of cannabis users and controls. EGJ—esophagogastric junction; LES—lower esophageal sphincter.

| Manometry Findings | Cannabis Users % (*n*) *n* = 21 | Controls % (*n*) *n* = 29 | *p* Value | Odds Ratio (OR) | 95% Confidence Interval (CI) |
|---|---|---|---|---|---|
| EGJ outflow obstruction | 19 (4) | 24 (7) | 0.66 | 0.7 | 0.2–2.9 |
| Impaired esophageal bolus clearance | 43 (9) | 17 (5) | 0.04 | 3.6 | 0.9–13.1 |
| Hypertensive LES | 29 (6) | 7 (2) | 0.04 | 5.4 | 0.9–30.1 |
| Normal | 9 (2) | 52 (15) | 0.002 | 0.1 | 0.01–0.05 |

## 3. Discussion

This study highlights the impact of cannabis on gastrointestinal physiology. This is the largest study to date assessing the gastrointestinal complaints of a cohort of patients with chronic cannabis use, who were evaluated in a gastroenterology outpatient clinic. Abdominal pain was the most common finding among chronic cannabis users, especially those who used cannabis daily. Visceral pain is thought to be regulated by the endocannabinoid system within the CNS. Mechanically evoked visceral pain models suggest that an increase in endocannabinoid levels alleviates visceral pain [2,13]. This finding was not observed in this study, suggesting that cannabinoids may not have the desired therapeutic analgesic effect demonstrated in animal model studies. This correlates with the findings of de Vries et al. whose study demonstrated no improvement in chronic abdominal pain with exogenous cannabinoid administration [10].

Significantly more cannabis users were African Americans compared to our randomized control group. This is contrary to national data which depicts that Caucasians encompass nearly three-fourths of those admitting to previous cannabis use [14]. Future research should investigate the effect of race and genetics on the symptomatology and clinical presentation of cannabis users.

There is limited data on esophageal manometry findings in patients who habitually use cannabis. Our study found significant findings of incomplete esophageal bolus clearance and hypertensive LES on manometry of cannabis users, which suggests that a potentially distinct esophageal manometry pattern may exist amongst this group. To date there are seven studies reporting a distinct pattern of a hypertensive LES, esophagogastric (EGJ) outflow obstruction and esophageal spastic peristalsis associated with opioid use prior to esophageal manometry [15–21]. None of these studies have accounted for cannabis usage as a potential confounder. This raises the question of whether cannabis, opioids, or a combination of both, impact esophageal motility accounting for the undeniable similarity in manometry findings.

Endoscopic findings of the upper GI tract of cannabis users has never been previously published. This study found that mucosal inflammatory changes in the upper GI tract on endoscopy and biopsy specimens of cannabis users were significantly more prevalent compared to non-cannabis users. On the contrary, cannabinoids have been implicated in mitigating inflammation and mucosal damage in GERD [22]. Toxins contained within cannabis, or the action of various exogenous cannabinoids on the ECS are possibly responsible for its pro-inflammatory properties. Future research examining the endoscopic and histologic effects of cannabis and its constituents are needed to better understand the pathophysiology of cannabis-induced mucosal injury.

Our study had several limitations. For one, it was conducted with a retrospective design utilizing an electronic medical record system with limited data inputting capabilities. As a result, we were unable to ascertain information such as reasons for cannabis use, exact duration of use, and exact timing of gastrointestinal complaints in relation to cannabis use. This information would have clarified and strengthened our study conclusions. Second, our control subjects were not matched for age, gender or race. We do not believe this is a strong limitation as none of our study end points, which included a variety of gastrointestinal symptoms, endoscopic findings, and manometric classifications, have been shown to be more prevalent in any specific age group, gender or race. In addition, our control subjects were randomly selected by our information technology department utilizing a 2:1 construct which mitigated potential bias imparted by our study design.

Another limitation of our study is that we did not collect all important data from our electronic medical record during our manual chart review. This information includes concomitant medications such as narcotics, motility agents and acid suppressive therapy, all which would have been important to control for. Evaluation of gastrointestinal emptying study data was also not collected in our study. These results may have complemented or explained the HREM findings in our study population. Furthermore, psychiatric disorders were not accounted for in this study. Previous studies have directly linked cannabis use to depression, anxiety and sleep quality which may impact the onset and/or the resolution of gastrointestinal symptoms [23,24]. Future studies should include all of this data in order to generate clear and strong conclusions.

## 4. Materials and Methods

This retrospective case-control study included patients evaluated at the Drexel University Gastroenterology outpatient clinic during 2006–2017. Patients aged 18 or older, who had at least two visit notes by at least two different providers within the Drexel outpatient electronic medical record (EMR) and cannabis use recorded in multiple visit notes, were included in the study. The Drexel University Information Technology (IT) department performed data mining of the Allscripts outpatient EMR to identify all patient charts which included the words "marijuana", "cannabis" or "THC" after approval (IRB ID: 1705005382; 16 May 2017) of the research protocol by the Drexel University Institutional Review Board (IRB).

The IT department also identified patients evaluated during this period without the above terms recorded in their charts. An IT analyst obtained a randomized list of controls by inputting these medical records numbers (MRNs) into data analysis software which arranged these MRNs in ascending order, then selected every 15th MRN thus obtaining randomly selected controls.

All charts of cases and controls were individually reviewed by the authors to verify documentation of cannabis use. Visit notes, endoscopy procedure reports, and manometry reports were manually reviewed to obtain study data parameters including main GI symptom, patient demographics, duration and frequency of cannabis use, and endoscopy and manometry findings. Patient information was stored in a secure, encrypted database.

Clinical and demographic characteristics of the cannabis and control cohorts were presented as frequencies (%) and proportions for categorical variables and means for continuous variables. Comparisons between categorical data such as ages, ethnicities, sex and symptomatology etc. were assessed using the Chi-square test or Fisher exact test where appropriate. A *p* value $\leq 0.05$ was considered statistically significant, and statistical tests were two-sided. All variables except frequency of cannabis use were dichotomized. The corresponding 95% confidence intervals were computed using the Clopper-Pearson exact method. Statistical analysis was performed using SPSS version 24.0 (SPSS, Chicago, IL, USA).

## 5. Conclusions

The effects of cannabis usage on the gastrointestinal tract is a relatively novel field of study, particularly involving human subjects. Our retrospective analysis of patients admitting to chronic

cannabis use provides a unique perspective on gastrointestinal symptoms and clinical manifestations in patients that use cannabis. Our study refutes conclusions derived from previous literature on this topic. Continued research efforts which focus on better understanding the pathophysiology and subsequent clinical consequences of cannabis on the gastrointestinal tract will be crucial for medical decision making in the future.

**Author Contributions:** Conceptualization, M.P., S.S. and A.A.; data curation, M.P., S.S. and A.A.; formal analysis, M.P., S.S. and A.A.; investigation, M.P., S.S. and A.A.; methodology, M.P., S.S. and A.A.; project administration, S.S. and A.A.; supervision, S.S. and A.A.; writing—original draft, M.P., S.S. and A.A.; writing—review and editing, M.P., S.S. and A.A.

**Funding:** This research received no external funding.

**Conflicts of Interest:** The authors declare no conflict of interest.

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
