# Peer review of "Cannabis Use in Patients Presenting to a Gastroenterology Clinic: Associations with Symptoms, Endoscopy Findings, and Esophageal Manometry"

_gastrointestdisord, doi:10.3390/gidisord1030025_

Round 1

Reviewer 1 Report

This is a good manuscript looking at symptoms, and test results in patients Using marijuana compared to those that do not. Some areas that should be improved upon.

Methods.  Were all patients seen in the clinic asked if they used marijuana?  Were only a subgroup asked?  if only a subgroup, this might lead to bias if patients with symptoms of N/V and abdominal pain were asked this more often

Aims. These can be clarified, as I find it a bit misleading on how the study was done.  This was a retrospective analysis of symptoms, test results, demographic items partitioned by if patients used MJ or did not.

Why were the patients using Marijuana?  Was it for there symptoms of abdominal pain, nausea, vomiting?  This might be a reason why there is this association.  Could marijuana be causing the symptoms.  Was MJ more recreational?  Was it medicinal marijuana?

The title can be improved upon. Why is abdominal pain, there but not heartburn, nausea/vomting.  Consider "Cannabis Use in Patients presenting to Gastroenterology Clinics: Associations with Symptoms, Endoscopy findings, and esophageal manometry.

Author Response

Point 1: Methods.  Were all patients seen in the clinic asked if they used marijuana?  Were only a subgroup asked?  if only a subgroup, this might lead to bias if patients with symptoms of N/V and abdominal pain were asked this more often

Response 1: Regarding our methods, it is expected that most patients, if not all, be assessed on their usage of drugs including marijuana and other potential toxic substances during their assessment in our outpatient healthcare system. Unfortunately, the possibility of bias is certainly present as we are not able to elucidate the quantity of patients that were assessed for marijuana use

Point 2: Aims. These can be clarified, as I find it a bit misleading on how the study was done.  This was a retrospective analysis of symptoms, test results, demographic items partitioned by if patients used MJ or did not.

Response 2: In reference to our aims, we detailed our approach and methodology as best as we could. I hope this met the standard of what the journal expects from its articles.

Point 3: Why were the patients using Marijuana?  Was it for there symptoms of abdominal pain, nausea, vomiting?  This might be a reason why there is this association.  Could marijuana be causing the symptoms.  Was MJ more recreational?  Was it medicinal marijuana?

Response 3: Most of the documentation in our electronic medical record fails to identify if marijuana usage was recreational or for medicinal purposes and this is certainly one of the limitations of our study. Specific reasoning as to why patients were using marijuana was also not indicated in most cases upon our chart review. One of the more interesting questions our study poses is “Does marijuana usage potentiate GI symptoms rather than suppress them?” and while we cannot definitively support its exacerbation of existing symptoms it is certainly an important point to be noted that may be further assessed via a prospective study in the future.

Point 4: The title can be improved upon. Why is abdominal pain, there but not heartburn, nausea/vomiting.  Consider "Cannabis Use in Patients presenting to Gastroenterology Clinics: Associations with Symptoms, Endoscopy findings, and esophageal manometry.

Response 4: We will modify our title to: Cannabis Use in Patients Presenting to a Gastroenterology Clinic: Associations with Symptoms, Endoscopy Findings, and Esophageal Manometry”

Reviewer 2 Report

This is a retrospective case-control study of patients attending a gastroenterology outpatient clinic over an 11 year period comparing cannabis user with non-cannabis users. 

1.      As in any such observational study, there is a significant risk of bias, particularly as the control and cannabis groups differ significantly in their demographics – the cannabis users being younger and more likely non-Caucasian in origin. The associations identified are therefore substantially weakened.  However this is partially mitigated by the ‘dose effect’ demonstrated for daily vs non-daily cannabis users.  It would be useful therefore to compare and show the demographics of these two subgroups as well (in table 3) – if they are more equivalent than the overall groups then this would serve to strengthen the associations further.

My other two points would be more difficult for the authors to achieve, although their data mining software may be able to generate this information.

2.      There may be other significant compounding factors, and medication use would also be useful to present – if accessible, as the demographic factors or associations of cannabis use may make individuals more or less likely to take prescribed medications on a regular basis. 

3.      A stronger study would have case-matched the controls rather than selecting them randomly. Given the number of available controls (only 1/15th were selected) and the power of the software used it should be possible to trawl the data for age, sex and racial origin and still generate sufficient randomised controls.

Overall the study is well presented and written and the findings are significant and warrant publication

Author Response

Point 1: As in any such observational study, there is a significant risk of bias, particularly as the control and cannabis groups differ significantly in their demographics – the cannabis users being younger and more likely non-Caucasian in origin. The associations identified are therefore substantially weakened.  However this is partially mitigated by the ‘dose effect’ demonstrated for daily vs non-daily cannabis users.  It would be useful therefore to compare and show the demographics of these two subgroups as well (in table 3) – if they are more equivalent than the overall groups then this would serve to strengthen the associations further

Response 1: Regarding comparing the demographics of cannabis vs non-cannabis users, we would have to do further in-depth statistical analysis to add to this section which would take us a considerable amount of time.

Point 2: There may be other significant compounding factors, and medication use would also be useful to present – if accessible, as the demographic factors or associations of cannabis use may make individuals more or less likely to take prescribed medications on a regular basis.

Response 2: While compounding factors such as concomitant medication usage would potentially strengthen our study, the data that we initially mined via assistance from our Information Technology department did not have this information available and it would be very difficult to obtain that information at this time. 

Point 3: A stronger study would have case-matched the controls rather than selecting them randomly. Given the number of available controls (only 1/15th were selected) and the power of the software used it should be possible to trawl the data for age, sex and racial origin and still generate sufficient randomised controls.

Response 3: The reasoning behind our selection of controls was to limit bias and obtain randomized controls to maintain an estimated 2:1 control: case ratio for our study. Regarding obtaining case-matched controls, we would have to repeat a request for data mining from our Information Technology department which would likely take a significant amount of time in addition to the extensive data compilation efforts with the new controls.

Reviewer 3 Report

General: The authors have identified an interesting research question. “Correlation between Mucosal Inflammation, Abdominal Pain, and Esophageal DYSFUNCTION in chronic Cannabis Users” is an interesting topic. The methods used are appropriate, and the presentation of the data is well performed. Minor corrections and grammar need to be improved before the article can be accepted.

The title is appropriate.

Abstract: look ok

Introduction, line 44 please add this sentence “Studies have shown that Cannabis use is common in patients with inflammatory bowel diseases to help relieve symptoms”. Please cite these articles Pubmed ID: 24366227 and PMID: 30290187

Materials and Methods section should be after the introduction section and not at the last.

In materials and methods please explain how cases and controls are selected and on what basis.

Line 144 to 150 can be included in a separate paragraph of Conclusion.

All in all, the analysis in this article is performed accurately and rigorously. Grammar needs to be rechecked throughout the manuscript.

Author Response

Point 1: Introduction, line 44 please add this sentence “Studies have shown that Cannabis use is common in patients with inflammatory bowel diseases to help relieve symptoms”. Please cite these articles Pubmed ID: 24366227 and PMID: 30290187

Response 1: Please see the revised manuscript Re: addition of the statement on cannabis usage in IBD patients and associated citations.

Point 2: Materials and Methods section should be after the introduction section and not at the last.

Response 2: The manuscript template document for the journal places the ‘Materials and Methods’ section prior to the conclusion which is what we followed. We can place this after the ‘Introduction’ if need be.

Point 3: In materials and methods please explain how cases and controls are selected and on what basis.

Response 3: Our cases were selected via assistance from our Information technology department, who data mined through our outpatient EMR to select patients who had been seen in our gastroenterology clinic, including only those charts with the keywords ‘marijuana’, ‘cannabis’ or ‘THC’ noted. Controls were selected by our IT department as well—these were patients without any notation of the keywords marijuana’, ‘cannabis’ or ‘THC’ in the chart that were evaluated in our gastroenterology clinic.  Of this cohort, every 15th chart was randomly selected (arranged as per ascending Medical Record number) to create a 2:1 ratio between controls and cases. The above is noted in our ‘Materials and Methods’ section. Please let us know if there are any other specifics that you would like to know regarding our methods.  

Point 4: Line 144 to 150 can be included in a separate paragraph of Conclusion.

Response 4: A ‘Conclusion’ section was included in the revised manuscript, with lines 145-151 placed under this heading.

Reviewer 4 Report

Correlation between Mucosal Inflammation, Abdominal Pain, and Esophageal DYSFUNCTION in  chronic Cannabis Users is a retrospective study that suggest that cannabis use may potentiate or fail to alleviate a variety of GI symptoms.

major revisions:

the randomization method is not appropriate. you should analyze control patients with the same symptoms of the cannabinoid users, and then compare the outcome.

you should stratify the cannabinoids user for consumption time (1 year? 10 years? months?), this will be a more interesting result than the comparation tou made with those controls.

Author Response

Point 1: the randomization method is not appropriate. you should analyze control patients with the same symptoms of the cannabinoid users, and then compare the outcome.

Response 1: Unfortunately, if we changed our randomization method to analyze control patients with the same symptoms as cannabis users we have to essentially re-do our study although this would potentially strengthen our findings.

Point 2: you should stratify the cannabinoids user for consumption time (1 year? 10 years? months?), this will be a more interesting result than the comparation tou made with those controls

Response 2: Given the shortcomings in chart documentation upon our review of the medical records we were able to only make the distinction between ‘daily’ and ‘non-daily’ cannabis users. A more thorough stratification of consumption time would potentially be a more intriguing comparison.

Round 2

Reviewer 4 Report

as suggested the authors answered the questions point by point 

The authors changed the text and now the study  is more understandable

Author Response

Thank you